# Inferring the Regulatory Network of miRNAs on Terpene Trilactone Biosynthesis Affected by Environmental Conditions

**DOI:** 10.3390/ijms242317002

**Published:** 2023-11-30

**Authors:** Ying Guo, Yongli Qi, Yangfan Feng, Yuting Yang, Liangjiao Xue, Yousry A. El-Kassaby, Guibin Wang, Fangfang Fu

**Affiliations:** 1State Key Laboratory of Tree Genetics and Breeding, Key Laboratory of Forest Genetics & Biotechnology of Ministry of Education, Co-Innovation Center for Sustainable Forestry in Southern China, Nanjing Forestry University, Nanjing 210037, China; qiyongli22@njfu.edu.cn (Y.Q.); yangfanfeng@njfu.edu.cn (Y.F.);; 2Department of Forest and Conservation Sciences, Faculty of Forestry, The University of British Columbia, Vancouver, BC V6T 1Z4, Canada; y.el-kassaby@ubc.ca

**Keywords:** *Ginkgo biloba* L., ginkgolides, bilobalide, miRNA, regulatory network, environment response

## Abstract

As a medicinal tree species, ginkgo (*Ginkgo biloba* L.) and terpene trilactones (TTLs) extracted from its leaves are the main pharmacologic activity constituents and important economic indicators of its value. The accumulation of TTLs is known to be affected by environmental stress, while the regulatory mechanism of environmental response mediated by microRNAs (miRNAs) at the post-transcriptional levels remains unclear. Here, we focused on grafted ginkgo grown in northwestern, southwestern, and eastern-central China and integrally analyzed RNA-seq and small RNA-seq high-throughput sequencing data as well as metabolomics data from leaf samples of ginkgo clones grown in natural environments. The content of bilobalide was highest among detected TTLs, and there was more than a twofold variation in the accumulation of bilobalide between growth conditions. Meanwhile, transcriptome analysis found significant differences in the expression of 19 TTL-related genes among ginkgo leaves from different environments. Small RNA sequencing and analysis showed that 62 of the 521 miRNAs identified were differentially expressed among different samples, especially the expression of miRN50, miR169h/i, and miR169e was susceptible to environmental changes. Further, we found that transcription factors (ERF, MYB, C3H, HD-ZIP, HSF, and NAC) and miRNAs (miR319e/f, miRN2, miRN54, miR157, miR185, and miRN188) could activate or inhibit the expression of TTL-related genes to participate in the regulation of terpene trilactones biosynthesis in ginkgo leaves by weighted gene co-regulatory network analysis. Our findings provide new insights into the understanding of the regulatory mechanism of TTL biosynthesis but also lay the foundation for ginkgo leaves’ medicinal value improvement under global change.

## 1. Introduction

*Ginkgo biloba* L., with a long history of more than 200 million years, is one of the oldest living plant species in nature [1,2]. Ginkgo trees have significant medical value as they contain many active ingredients, such as terpene trilactones (TTLs), flavonoids, and ascorbic acid [3]. TTLs, a unique terpenoid in Ginkgo, play an essential role in treating cardiovascular and neurological diseases [4]. TTLs are diterpenes with a cage skeleton consisting of six five-membered rings that differ only in the number and position of hydroxyl groups [5]. To date, more than ten ginkgolides and bilobalide have been isolated from ginkgo leaves and root barks [6], while the content of TTL components in ginkgo leaves is too low to sustain the market demands. In the past 20 years, several attempts have been made to improve TTLs contents. For example, Crimmins et al. succeeded in synthesizing ginkgolide B through chemical routes [7]; however, this method was not amenable for commercial-scale production [8]. In vitro cultures were also attempted to promote ginkgolide and bilobalide production, but the method has shortcomings, such as a low yield and long cultivation period. Notably, the synthesis of TTLs is affected by many factors, including internal developmental genetic circuits and external environmental factors [9], which can be integrated as tools for increasing their accumulation.

It was shown that the methylerythritol 4-phosphate (MEP) pathway is mainly responsible for the biosynthesis of ginkgolides [10], while bilobalide synthesis was thought to take place through the cytosolic mevalonate (MVA) pathway [11], with cross-talks between these two pathways. Environmental stress could increase the content of TTLs in ginkgo by inducing the expression of TTL-related genes. For instance, under UV exposure for 48 h, there was a 10% surge in TTL content, attributed to the significant upregulation of *GbDXS* (1-deoxy-D-xylulose 5-phosphate synthase), *GbGGPS* (geranylgeranyl diphosphate synthase), and *GbLPS* (levopimaradiene synthase) genes. Similarly, cold stress for a period of 8 days led to a 14.5% boost in TTL content, again due to the significant increase in the expression levels of *GbDXS*, *GbGGPS*, *GbLPS*, and *GbMVD* genes [12]. While studies related to TTL biosynthetic pathways have made significant strides, the detailed molecular mechanisms underpinning TTL metabolism in responses to varying environmental conditions remain to be fully elucidated.

MicroRNAs (miRNAs) play a crucial role in mediating fundamental processes for plant survival in response to environment stress. MiRNAs are post-transcriptional regulators that target genes to negatively regulate them by cleavage and/or translational inhibition [13]. A few miRNAs have been identified to participate in TTL metabolism regulation. For instance, 25 miRNAs potentially played a vital role in TTL accumulation in ginkgo leaves by targeting key transcription factor genes [14]; Singh et al. found that miR156 could regulate the TTL metabolism pathway in *Mentha* spp. by targeting the *DXS* gene [15], and miR5021 could regulate this pathway by targeting *DXS*, *GPS*, and *GGPS* genes in *Curcuma longa* [16]. As endogenous gene modulators, moreover, several core miRNAs have been confirmed to participate in the biological processes of plant environmental adaptation. For example, miR396c/d could inhibit the expression of *GRF1* in maize to enhance drought stress tolerance [17]; the overexpression of miR319b regulated *OsPCF6* and *OsTCP21* to increase proline accumulation, thereby improving the cold tolerance ability of rice [18]; and the study in *Panicum virgatum* L. found that the down-regulation of miR166 expression could mitigate salt stress [19]. Thus, deciphering the complex molecular networks regulated by miRNAs is essential to improve TTLs synthesis and accumulation in response to changing environments.

In this study, we integrated RNA-seq and small RNA-seq high-throughput sequencing data as well as metabolomics data from leaf samples of ginkgo clones grown in different environments. Our specific objectives were to (1) investigate the response of TTL accumulation to different environments; (2) annotate miRNAs at the ginkgo genome-wide level and identify miRNAs involved in the regulation of TTL metabolism therein; and (3) reveal the regulatory mechanism mediated by miRNAs in response to TTL metabolism in different environments. This study is expected to provide new insights into the regulatory mechanism of TTL biosynthesis and to lay a theoretical foundation for ginkgo leaves’ medicinal value improvement.

## 2. Results

### 2.1. Changes in TTLs Content under Different Environments

Four TTLs contents (ginkgolide A, B, C, and bilobalide) in ginkgo leaves from three different habitats were determined by HPLC-ELSD, and the molecular structural formula of TTLs is shown in Figure 1. The content of bilobalide was highest among the four TTLs, and there were significant differences (*p* < 0.05) in the accumulation of bilobalide under the three environments. The content of ginkgolide C decreased gradually from S1 to S3, and significant differences were found between S1 and S3 samples, while there were no significant differences in the content of ginkgolide A and B among leaf samples from the three different environments. Collectively, the total contents of the four TTLs in S1 (ranging from 6.15 to 7.95 mg/g) were significantly higher than the other two sites (S2 and S3), suggesting that TTLs accumulation was affected by environmental conditions (Figure 1).

The metabolome database was collected from our previous study [20]. There were 13,742 substances detected from the mass spectrums of nine selected ginkgo samples, including 8 substances in the terpenoid backbone biosynthesis pathway, such as Acetyl-CoA, DXP, CDP-ME, CDP-MEP, ME-cPP, HMBPP, Isopentenyl-PP, and FPP (Figure 2). Compared with S2 and S3 samples, the abundance of CDP-ME, HMBPP, and Isopentenyl-PP was higher in S1 samples, which were essential precursors for the biosynthesis of TTL (Figure 2).

### 2.2. Analysis of Environment-Responsive mRNAs

The 25,328 genes that could be expressed (TPM > 0) were found in 27,832 genes annotated in the reference genome via transcriptome analysis (NCBI BioProject number: PRJNA649066). We found a total of 8496 differentially expressed genes (DEGs) between the S1 and S3 samples, which was approximately twice the number of DEGs between S1 and S2 samples (Appendix A). Next, we carried out GO and KEGG enrichment analysis, and 7049 and 863 DEGs were enriched in GO terms (Appendix A) and KEGG pathways, respectively (Appendix A). For example, larger numbers of DEGs were observed in GO terms of the cellular polysaccharide metabolic process, fatty acid biosynthetic process, and external encapsulating structure organization, and some DEGs were enriched in the biosynthesis of amino acids, carbon metabolism, and glycolysis or gluconeogenesis pathways. Moreover, we identified 56 structural-related genes on the terpenoid backbone biosynthesis pathway, 19 of which were differentially expressed in the three environments, such as *GbGGPS*, *GbHMGR*, and *GbHDR* (Figure 2).

### 2.3. Annotation of miRNAs and Their Target Genes

To explore the response of miRNAs to environmental changes, we first identified ginkgo miRNA loci at the genome-wide level using nine sRNA-seq datasets (NCBI BioProject number: PRJNA903548) and 28 public ginkgo sRNA libraries. Here, 521 miRNAs (from 337 miRNA families) were annotated, including 206 known and 315 novel miRNAs (Figure 3A and Appendix A). As shown in Figure 3B, the size of miRNAs ranged between 20 nt and 24 nt, with 21 nt having the highest proportion of identified miRNAs (77.5%). The first nucleotide of the 5′ end of mature miRNAs had a significant bias towards U (Figure 3C). Additionally, we found that 206 known miRNAs came from 54 conserved miRNA families. In summary, the family analysis showed that the detected miRNA family miR11534 was the most abundant with 18 members, followed by miR396 with 14 members and miR169 with 13 members (Figure 3D). Of the remaining miRNA families, 278 were represented by only one family member. Further, we predicted the targeted regulation of 20,446 genes by 521 miRNAs using psRNATarget software (http://plantgrn.noble.org/psRNATarget/ accessed on 22 September 2023), covering about 73% of the total genes (Appendix A). The genome-wide annotation of miRNAs in ginkgo could provide support for further exploring the diversity and regulatory functions of miRNAs in gymnosperms.

Among the 521 miRNAs annotated, 62 differentially expressed miRNAs (DEMs) were found among three environments (Figure 4A and Appendix A). Consistently, more DEMs were identified between S1 and S3 samples, with the largest differences in environmental conditions. As shown in the Venn diagram (Figure 4B), three DEMs (gbi-miRN50, gbi-miR169h/i, and gbi-miR169e) were shared in the three comparison groups. There are 6346 genes predicted to be the target genes of 62 DEMs. Furthermore, we performed gene ontology (Appendix A) and KEGG pathway (Appendix A) enrichment analysis to explore the biological functions of target genes. Our gene ontology analysis showed that DEM target genes were involved in the biological processes of copper ion transport, transition metal ion transmembrane transporter activity, and ADP binding. Remarkably, we found that some target genes of DEMs were enriched in a few of the secondary metabolite biosynthesis pathways, such as flavonoid and terpenoid backbone biosynthetic pathways (Appendix A). Five target genes of nine DEMs were related to TTL biosynthesis (Figure 4C), such as *GbAACT* (chr4.1506), *GbPMVK* (chr12.788), *GbHDS* (chr12.1656), *GbHMGR* (chr9.1771), and *GbGGPS* (chr8.380).

### 2.4. Construction of TTL-Related Gene Regulatory Network

Integrating omics datasets, the regulation mechanism of the TTL biosynthesis was explored through weighted gene co-expression network analysis (WGCNA). As shown in Figure 5A, thirty-two modules with similar expression patterns were identified (Appendix A), and the number of genes in modules ranged from 4 to 8781 (Appendix A). There were four modules with more than one thousand genes, such as Yellow, Brown, Blue, and Turquoise modules. We identified 634 transcription factor (TF) coding genes in all modules by using the PlantTFDB online website (Appendix A). To find out the modules related to TTL biosynthesis, we quantified the correlation between each module and TTL traits. Notably, the Blue module eigengene was positively related to the content of various TTL and their precursors (r > 0.8, *p* < 0.05), such as CDP-ME, HMBPP, bilobalide, and total TTL (Figure 5B). In the Blue module, we found 12 genes encoding enzymes in the TTL biosynthesis pathway and extracted 30 genes closely related to their expression from the module based on weight values. In order to find the key regulators, we identified TF coding genes from the extracted gene set and retrieved miRNAs targeting these structural genes and TF coding genes. Finally, we constructed a sub-network based on the correlation coefficients of miRNA, TF gene, and structural gene expression. As shown in Figure 5C, the expression of 10 genes encoding TTL-related enzymes (*GbMVD*, *GbHMGS*, *GbHMGR*s, and *GbGGPS*s) and 18 TF encoding genes presented a certain degree of positive correlation, 13 of which were ERF encoding genes. Meanwhile, we observed the negative regulatory relationships between six miRNAs and their targets (one TF encoding gene and five structural genes), such as miR319e/f, miRN2, miRN54, miR157, miR185, and miRN188. Hence, we suggested that these TFs and miRNAs could play an important role in regulating TTL biosynthesis in ginkgo leaves, activating or inhibiting the expression of key enzyme genes. The qRT-PCR analysis of ten TTL-related genes showed the same expression patterns as the RNA sequencing results, albeit with some subtle differences, confirming the reliability of these results (|r > 0.65|, *p* < 0.005; Appendix A). Furthermore, 37 Cytochrome P450 (CYP450) genes were found in the Blue module, of which the expression of 36 CYP450 family members was significantly correlated with that of TTL-related structural genes (r > 0.6, *p* < 0.05), indicating that *GbCYP450* may be involved in regulating the biosynthesis of terpene trilactone.

## 3. Discussion

### 3.1. Effects of Environmental Condition on TTLs Accumulation in Ginkgo

Ginkgolides (GA, GB, GC, GM, GJ, GP, GQ, GK, GL, GN) and bilobalide (BB) are TTLs isolated from ginkgo leaf and its root bark. TTLs have a high medicinal value and play an essential role in curing cardiovascular diseases [4]. As an important secondary metabolite, TTLs accumulation could be influenced by ecological factors such as water, temperature, and light [21]. This study determined GA, GB, and GC and BB contents in leaf samples from three different environments. We found that the total contents of four TTLs in S1 samples were obviously higher than the other two samples (Figure 1). According to the analysis of climate data, S1 has the highest precipitation (S1, 316.3 mm; S2, 101.2 mm; S3, 4.0 mm) and the lowest evapotranspiration (107.7, 118.6, 199.5 mm) in sampling month (Appendix A). A previous study found that moderate drought conditions could promote TTLs biosynthesis in ginkgo leaves [9], whose accumulation could eliminate reactive oxygen species (ROS) during abiotic stress, blocking oxidation and protecting plant growth [22]. It is speculated that moderate drought could induce additional TTLs accumulation in S1 samples by regulating the gene expression of the abscisic acid biosynthesis enzyme, transcription factors, and TTL biosynthesis-related enzyme [23]. A previous study showed that TTLs in ginkgo leaves treated with a high temperature increased compared with the control group [24], while the average monthly maximum temperature of S2 is 4 °C, which is 6 °C lower than that of S1 and S3, respectively (Appendix A). Wang et al. indicated that there was an optimum light intensity in a specific light range, and when light intensity was higher or lower than this optimum, a pronounced decrease in TTLs content was observed in ginkgo leaves [25]. The average sunlight hour in August of S3 is more than 9 h (9.16 h), while that of S1 (5.76 h) and S3 (5.91 h) is less than 6 h (Appendix A). Excessive radiation may be a substantial reason for the decreased TTLs accumulation in S3 samples. According to our previous research, the test sites (S1, S2, and S3) were located in high-, medium-, and low-suitability habitats, respectively [20]. Here, we suggest that priority should be given to establishing the Ginkgo plantations in high-suitability habitats to maximize TTLs production in harvested leaves.

### 3.2. Key Genes Regulating TTL Biosynthesis and Accumulation

Combined with previous plants’ TTL metabolism studies [5] and mRNA functional annotation, we found 56 structural genes involved in the TTL metabolism pathway, including 19 DEGs among samples from different environments (Figure 2). Focused on the connection between gene expression and metabolite abundance, we found that *GbHMGR*s were highly expressed in the S1 samples, which had a consistent pattern of downstream Isopentenyl-PP content. In addition, *GbHDS* was highly expressed in S1, thus resulting in a high abundance of HMBPP. GGPS is a vital enzyme associated with the biosynthesis of ginkgolides in ginkgo, and the expression of *GbGGPS*s was enhanced in S1, which may increase the abundance of precursors for the biosynthesis of ginkgolides (Figure 1 and Figure 2).

In ginkgo leaves, the expression of TTL-related genes was reported to be regulated by transcription factors from the bHLH, WRKY, and AP2 families [14]. In this study, through WGCNA, we identified 18 TF coding genes closely related to the TTL metabolism pathway (Figure 5C). Among these TF genes, 12 genes from the ERF family played a vital role in regulating the largest number of structural genes in this pathway. The ERF family is one of the AP2/ERF family factors found to regulate terpene biosynthesis in many plants, such as *Artemisia annua*, *Citrus sinensis*, and *Zea mays* [26,27,28]. A previous study of *Litsea cubeba* showed that *LcERF19* could enhance *LcTPS42* expression to improve geranial and neral biosynthesis [29]. The study of *Catharanthus roseus* found that AP2/ERF transcription factor coding gene, *ORCA3*, and its regulatory factor *CrMYC2* play a key role in terpenoid indole alkaloids biosynthesis [30]. In addition, it was found that *MYB* and *NAC* genes play important roles in regulating TTL biosynthesis, which is consistent with previous studies [31].

### 3.3. MicroRNA Functions in Environmental Stress Responses

In this study, we identified miRNA using a recently published high-quality reference genome to obtain an accurate annotation of miRNAs in ginkgo [32]. In addition, our miRNA method identification was robust and comprehensive. First, we selected 37 sRNA sequencing samples from different organs of ginkgo, ensuring that the miRNA library we built was not affected by sampling bias. Second, we used the ShortStack program to identify miRNAs with a rigorous set of structure- and expression-based parameter criteria. We identified 521 miRNAs belonging to 337 miRNA families (Figure 3A).

The adaptive response of plants to sudden environmental changes is a complex phenomenon, and an increasing number of studies have revealed that miRNAs are able to regulate a new gene expression program to help restore homeostasis [33]. Here, we found 62 DEMs among the three environments; in particular, the expression of miRN50, miR169h/i, and miR169e was vulnerable to environmental changes (Figure 4B). The miR169 family is a large and conserved family in plants, whose members are thought to be environmental-responsive miRNAs. The response of miR169 family members to drought stress was found in a series of studies, such as maize, *Echinacea purpurea,* and *Phaseolus vulgaris* [34,35,36]. An *Arabidopsis* study found that miR169, as an ambient temperature-responsive microRNA, played a vital role in stress responses and the floral transition [37].

As shown in Figure 4C, we found that the differentially expressed miR169l.1 directly binds to the promoter of *PMVK* (chr12.788), which was probably involved in the regulation of TTL biosynthesis. In the co-expression regulation sub-network, we identified four novel miRNAs and one known miR319 associated with the TTL metabolism pathway by targeting structural genes (Figure 5C). Previous studies found that miR319 played an important role in plant development and stress responses. Sun et al. found that miR319 controlled secondary cell wall formation during plant development by regulating TCP4 TF [38]. Furthermore, miR319 is highly correlated with plant drought resistance [39]. Interestingly, we found that miR319 differentially expressed in three environments can negatively regulate the expression of its target gene *HMGR* (DEG), which may further affect the synthesis and accumulation of downstream TTLs (Figure 4C and Figure 5C).

## 4. Materials and Methods

### 4.1. Plant Materials

Ginkgo leaves were collected from 2-year-old clonally propagated (grafted) ginkgo trees grown in three different test sites. We implemented randomized block experiments, where each test site was set up with three blocks, with 20 ginkgo seedlings planted in each block and a row spacing of 40 × 60 cm. The first site (S1) is located in Jiangsu Province, central China, with a longitude and latitude of 34.21 °N and 117.58 °E, respectively, with a mean annual temperature (MAT) of 14.5 °C and mean annual precipitation (MAP) of 845 mm. The second site (S2) is located in Yunnan Province, southern China (latitude: 25.52 °N, longitude: 103.58 °E), characterized by a warm and humid environment (MAT: 14.1 °C, MAP: 1067 mm). And the third site (S3) is located in Xinjiang Uygur Autonomous region, northwest of China with typical continental semi-arid climate characteristics (latitude: 43.41 °N, longitude: 81.11 °E; MAT: 5.2 °C, MAP: 331 mm). Further, we obtained daily climate data of the three test sites in the sampling month from the National Meteorological Science Data Center [40], including ten climatic indicators such as ground temperature, precipitation, and sunlight hours (Appendix A). At the three test sites, three vigorous ginkgo trees with a similar level of growth were randomly selected from each block (three biological replications) to collect 3–7 leaves at the upper end of the main branch for the sequencing of transcriptome, small RNA, and metabolome, and then 10 leaves were randomly collected for TTL content determination.

### 4.2. Extraction and Determination of TTL-Related Components

The extraction and determination of TTLs in ginkgo leaves were carried out according to the method in the People’s Republic of China protocol [41]. The fresh leaves were first dried at 105 °C for 15 min and then dried further (70 °C, 48 h), crushed using a micro-high-speed universal pulverizer (JC-FW200), and screened using a 100-mesh sieve. We wrapped 1.0 g of dry leaf powder in filter paper and immersed it in a Soxhlet extractor containing 90 mL of petroleum ether, and this was refluxed for one hour to remove impurities at 70 °C; the leaf powder was then immersed in 70 mL of methanol and refluxed for 6 h at the same temperature. Then, the extract was evaporated on the Yarong rotary evaporator (Shanghai, China) under the conditions of the relative vacuum of 95 kPa, heating water bath temperature of 65 °C, cooling medium temperature of 20 °C, and rotating speed of 60 r/min, and the residue was dissolved with 10 mL methanol. After that, the eluate was sonicated for 30 min at 50 °C with 300 W power and 40 KHz frequency using Skymen ultrasonic cleaner (Shenzhen, China), and then 5 mL of the supernatant was withdrawn and filtered through an aluminum oxide column. Next, the aluminum oxide column was washed using 25 mL methanol and the eluate was evaporated on a rotary evaporator, after which the residue was dissolved with 5 mL methanol. Then, the tube containing the eluent was sonicated for 30 min after adding 4.5 mL of deionized water to the tube. After cooling to room temperature, we diluted the eluent with a 50 mL volumetric flask of methanol to a constant volume and pipetted 1 mL of the solution for TTL content determination using high-performance liquid chromatography (HPLC). TTLs determination was performed using an evaporative light scattering detector (ELSD) at ambient temperature and carrier gas pressure 40 psi. The mobile phase was tetrahydrofuran solution (10%), methanol (25%), and deionized water (65%) at 1.0 mL·min^−1^. Standard ginkgolide (GA, GB, and GC) and bilobalide (BB) were provided by Shanghai yuanye Bio-Technology Co., Ltd., Shanghai, China. The HPLC chromatograms of the four standards are shown in Appendix A.

According to the previously reported method [20], metabolites were extracted from 9 freeze-dried samples of ginkgo leaves (three biological replicates) from different experimental sites, respectively. All of the samples’ extracts were mixed as the quality control (QC) samples, which could monitor deviations of the analytical results and evaluate potential errors from analytical instruments. An UHPLC system (1290, Agilent Technologies, Santa Clara, CA, USA) was used for the LC–MS/MS analyses. The peak intensities were batch normalized to the total spectral intensity, and the mass spectrum of the QC samples is shown in Appendix A. The identification of metabolites is based on the exact molecular formula (molecular formula error < 20 ppm). For compound identification, the XCMS online program (The Scripps Research Institute, San Diego, CA, USA) with OSISMMS (version 1.0, Dalian Chem Data Solution Information Technology Co., Ltd., Dalian, China) was used for peak annotation. We mapped the identified metabolites to the Kyoto Encyclopedia of Genes and Genomes (KEGG) databases [42] to determine the TTL precursors in the terpenoid backbone biosynthesis pathway (ko00900).

### 4.3. RNA Sequencing (RNA-seq) and Analysis

Total RNA was extracted from nine freeze-dried leaf samples using the Trizol reagent kit (Invitrogen, Carlsbad, CA, USA) following the manufacturer’s protocol. Follow the previous method for library preparation and transcriptional sequencing [20]. To obtain high-quality reads, adaptors of raw reads were removed by Trimmomatic (version 0.39) [40]. The processed reads were mapped to the ginkgo reference genome using Bowtie2 (version 2.3.0) [32,43]. To quantify mRNA expression, the TPM (Transcripts per million) value of each transcription region was calculated using RSEM (version 1.3.3) [44]. Differentially expressed genes (DEGs) were identified using DESeq2 [45] with a threshold of FDR < 0.05 and FC (fold change) > 1.5. All mRNA sequences were compared with the Swiss-Prot [46], Gene Ontology (GO) [47], and KEGG databases [42] using BLASTx software (https://blast.ncbi.nlm.nih.gov/ accessed on 22 September 2023) to perform gene functional analysis. Genes encoding key enzymes in TTL metabolism pathway were extracted according to KEGG annotation. Then, genes encoding transcription factors (TFs) were identified using PlantTFDB [48].

### 4.4. Small RNA Sequencing (sRNA-seq) and Analysis

Consistent with RNA-seq, sRNA libraries of nine freeze-dried leaf samples were constructed through TruSeq Small RNA Sample Prep Kits and sequenced using the Illumina Hiseq2000/2500 sequencer (50-bp single-end reads). An additional 28 small RNA data of ginkgo were downloaded from NCBI Sequence Read Archive (SRA) for analysis (Appendix A). Referring to the previous method for identifying miRNAs in poplar [49], we used the ShortStack program (version 3.3.3) [50] to comprehensively annotate the miRNA loci in the ginkgo reference genome [32]. In short, we sequentially used Cutadapt software (version 2.10) [51], Bowtie software (version 1.3.0) [52], ShortStack program, and PatMaN software (version 1.2) [53] for sequencing data filtering, alignment, and miRNA recognition. Additionally, the quantification of miRNA expression levels was based on TPM. Differentially expressed miRNAs (DEMs) were identified using R package DESeq2 [45] with the threshold of FDR < 0.05 and FC > 1.5. Mature miRNA and transcript sequences were submitted to the online tool psRNATarget to conduct target prediction with default parameters (maximum cutoff of score = 5) [54].

### 4.5. Constructing Co-Expression Network

After the filtration of unexpressed genes (TPM = 0) from the nine RNA samples, 25,328 expressed genes were preserved for weighted gene co-regulatory network analysis (WGCNA) [55]. The dynamic decision-making tree and adjacency matrix method were used to identify similar modules using R package WGCNA with the following parameters: min module size = 30; merge cut height = 0.25. In addition, a correlation analysis for the content of TTL-related components and epigengene was performed. In the module highly related to TTL accumulation, TTL-related structural genes were used as the hub genes to extract 30 genes close to them based on the weight value. We identified the transcription factor coding genes in the extracted gene set to select TF genes with a greater than 0.8 correlation with structural genes. Next, we retrieved miRNAs targeting these structural and TF coding genes, and miRNAs with a negative correlation between their expression (r < −0.6) were retained. Finally, a co-expression regulation sub-network was constructed based on the correlation among miRNAs, TF genes, and TTL-related structural genes, which were visualized using CytoScape (version 3.7.1) [56].

### 4.6. Quantitative Real-Time PCR (qRT-PCR) Analysis

Ten genes with a high expression involved in the regulation of TTL synthesis in ginkgo leaves were chosen for validation by qRT-PCR. Following the manufacturer’s guidelines, cDNA was synthesized using MonScript RTIII All-in-One Mix with dsDNase kits (Monad, Suzhou, China). Then, the qRT-PCR analysis was conducted using cDNA templated on an Applied Biosystems™ 7500 Real-Time PCR System. Primer Premier 5 was used to design the primers, and the sequences are listed in Appendix A. The glyceraldehyde 3-phosphate dehydrogenase (*GAPDH*) gene served as a reference gene. Each sample consisted of three biological replicates and three independent technical replicates. The relative gene expression was calculated using the 2^−ΔΔC^t^^ approach [57].

## 5. Conclusions

Ginkgo trees have the ability to biosynthesize diverse metabolites as defense substances to protect their long-term survival under complex environmental conditions. Here, we observed notable differences in the TTL content of ginkgo leaves from different environments, suggesting that some environmental factors are conducive to (moderate drought and high temperature) or not conducive to (excessive radiation) TTL biosynthesis and accumulation. Using the constructed regulatory network of miRNA-mRNA, we found several transcription factors and microRNAs to be essential regulators participating in the TTL biosynthesis pathway, which could provide new targets to enhance TTL accumulation at the molecular level. In a future study, we will investigate the relationship between miRNAs and target genes using a dual luciferase reporter assay to offer a new approach to improve TTLs accumulation in ginkgo leaves.

## Figures and Tables

**Figure 1 ijms-24-17002-f001:**
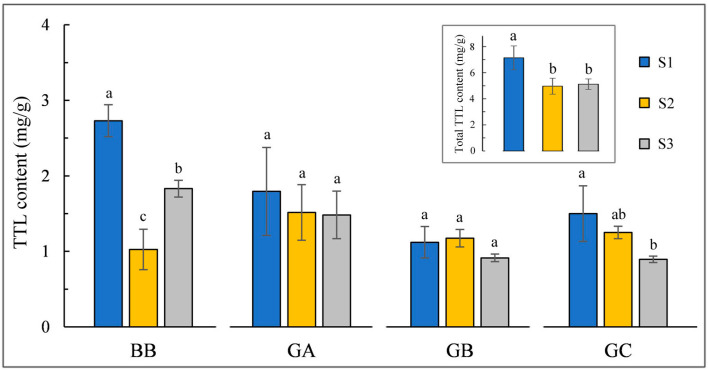
Variation in TTL content in ginkgo leaves from three test sites (S1–S3). The total TTL content is the sum of bilobalide (BB), ginkgolide A (GA), ginkgolide B (GB), and ginkgolide C (GC) contents. Error bars represent standard deviations, and small letters indicate the significant difference (*p* < 0.05).

**Figure 2 ijms-24-17002-f002:**
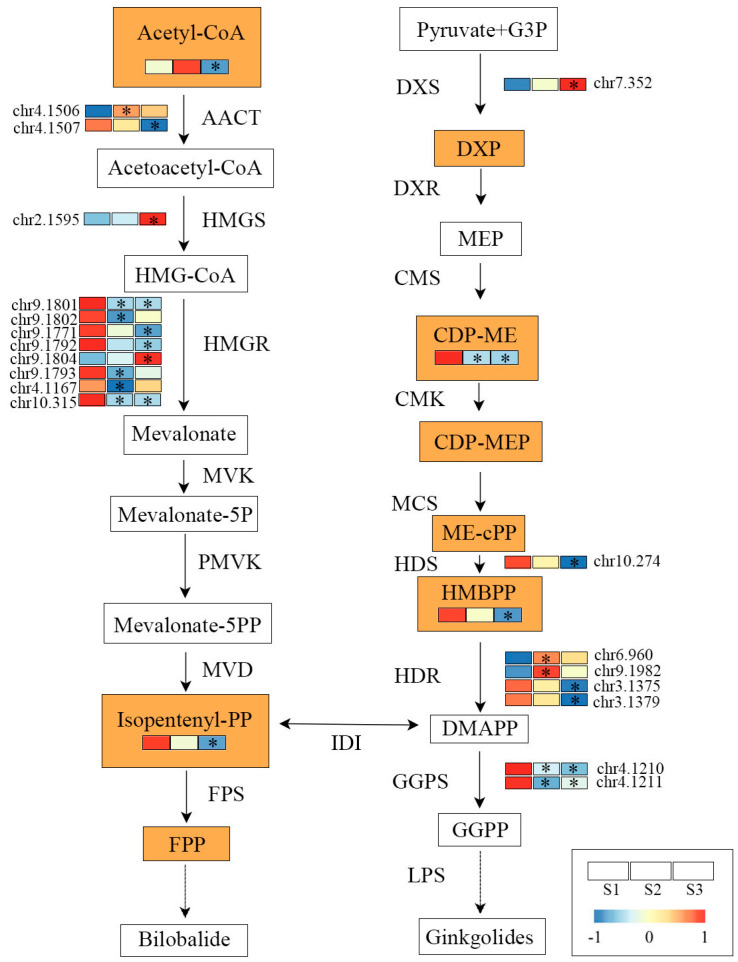
The pathway related to TTL metabolism in ginkgo and the heatmap showing the expression patterns of differentially expressed genes (DEGs) involved in TTL metabolism. The color change from red to blue indicates a gradual decrease in gene expression. The stars (*) in rectangular showed that there were significant differences between S2/S3 and S1 samples (*p* < 0.05). Enzyme names are abbreviated as follows: Acetyl-CoA C-acetyltransferase (AACT); Hydroxymethylglutaryl-CoA synthase (HMGS); Hydroxymethylglutaryl-CoA reductase (HMGR); Mevalonate kinase (MVK); Phosphomevalonate kinase (PMVK); Diphosphomevalonate decarboxylase (MVD); 1-deoxy-D-xylulose-5-phosphate synthase (DXS); 1-deoxy-D-xylulose-5-phosphate reductoisomerase (DXR); 2-C-methyl-D-erythritol 4-phosphate cytidylyltransferase (CMS); 4-diphosphocytidyl-2-C-methyl-D-erythritol kinase (CMK); 2-C-methy-D-erythritol 2,4-cyclodiphosphate synthase (MCS); (E)-4-hydroxy-3-methylbut-2-enyl-diphosphate synthase (HDS); 4-hydroxy-3-methylbut-2-en-1-yl diphosphate reductase (HDR); Farnesyl diphosphate synthase (FPS); Geranylgeranyl diphosphate synthase (GGPS); Isopentenyl-diphosphate Delta-isomerase (IDI); Levopimaradiene synthase (LPS).

**Figure 3 ijms-24-17002-f003:**
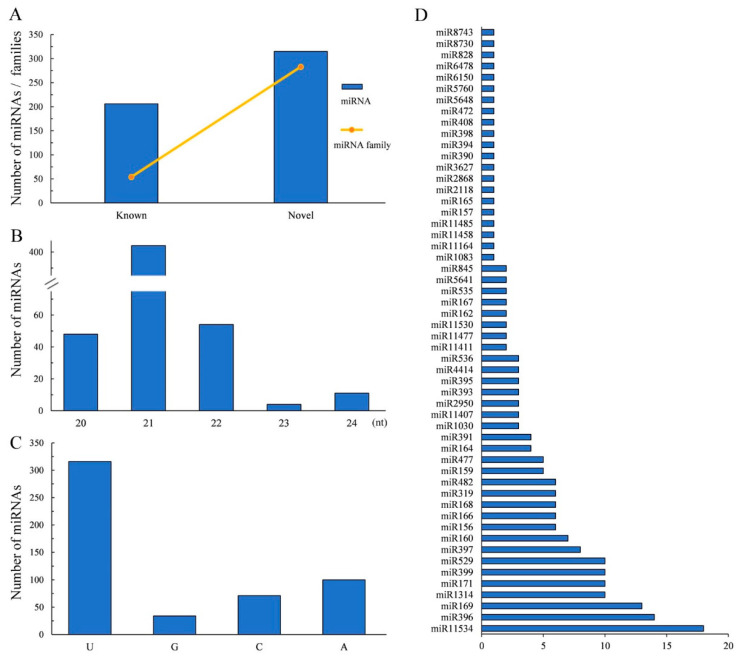
Annotation of miRNAs in ginkgo genome. (**A**) Number of miRNAs and miRNA families, (**B**) length distribution of identified miRNAs, (**C**) distribution of their 5′ end bases, and (**D**) number of conserved miRNA family members.

**Figure 4 ijms-24-17002-f004:**
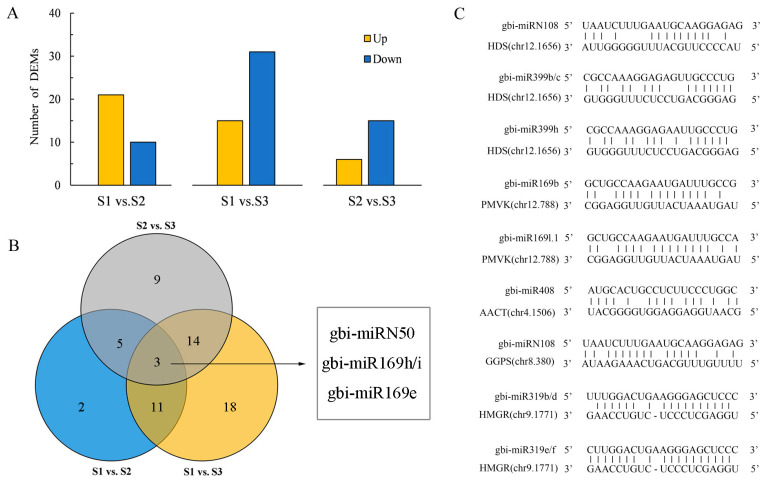
Analysis of the differentially expressed miRNAs (DEMs) among the samples from three environments. (**A**) The number of up- and down-regulated DEMs. Yellow rectangles represent up-regulated DEMs; blue rectangles represent down-regulated DEMs, (**B**) Venn plot shows the overlap of all identified DEMs in pairwise comparison. The blue circle represents the DEMs between S1 and S2 samples; yellow circle represents the DEMs between S1 and S3 samples; and gray circle represents the DEMs between S2 and S3 samples. The number represents the number of identified DEMs, (**C**) Prediction of DEM target sites of genes involved in TTL metabolism.

**Figure 5 ijms-24-17002-f005:**
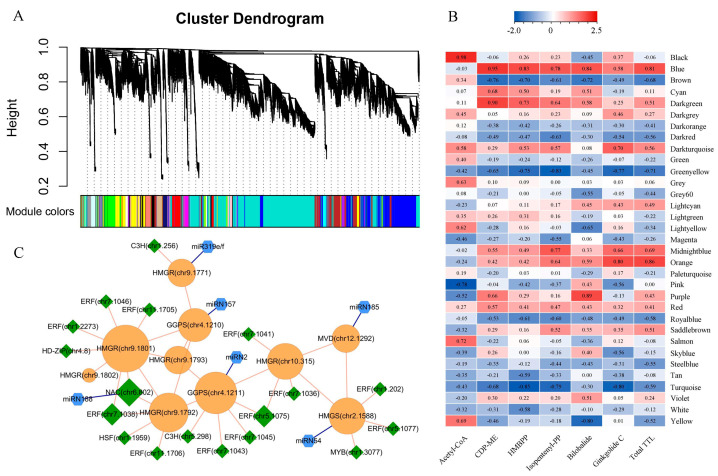
Construction of TTL-related gene regulatory network. (**A**) Clustering dendrogram showing modules of expressed genes identified by the WGCNA, (**B**) heatmap showing correlation of module eigengene and TTL metabolite content, (**C**) the sub−network of miRNAs and genes related to TTL metabolism. Orange circular: structural genes involved in TTL metabolism pathway; green quadrangle: TF coding genes; blue hexagon: miRNAs. The size of the shapes represents the level of connectivity. The larger the size, the higher the connectivity. The orange and blue lines represent positive and negative correlations, respectively.

## Data Availability

The sRNA sequencing data generated as part of this study are deposited in the NCBI Sequence Read Archive (SRA), accession BioProject number PRJNA903548.

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
