# Peer review of "Inferring the Regulatory Network of miRNAs on Terpene Trilactone Biosynthesis Affected by Environmental Conditions"

_ijms, 2023, doi:10.3390/ijms242317002_

Round 1
Reviewer 1 Report (Previous Reviewer 2)
Comments and Suggestions for Authors
Find attached my comments to improve the manuscript.

Comments on the Quality of English Languageminor revision
Author Response
RESPONSE TO REVIEWERS
Reviewer #1:
- Line 35: Add current references in ginkgo leaves (2023)
Response: We have cited the latest published relevant papers (L: 497-498).
- Line 94: Critically discuss the need for this research and the novelty
Response: Thanks for your suggestion and added sentences to discuss the novelty of this study (L: 79-81).
- Line 108: What type of hlplc are you talking about? Be specific
Response: Corrected (L: 95 and L: 342-344).
- Line 246: Talk more about the chemistry behind your results.
Response: Thanks for your suggestion and we added the discussion (L: 233-235).
- Line 336: Why dried at 70C for 48 hours since TTL is vulnerable to temperature and this will not give the actual TTL in GBL? Why not freeze-dried?
Response: Before drying (70 °C, 48 hours), fresh leaves are first dried at 105 °C for 15 min to terminate enzyme activity, which is a common method in production practice (L: 325).
- Line 337 Crushed using what?
Response: Crushed by a micro-high-speed universal pulverizer (JC-FW200) (L: 337).
- Line 339: What impurities? chlorophyll.?
Response: Lipid compounds.
- Line 359: "metabolite extract of each freeze-" where is the sample preparation for that?
Response: We have added the sentence to describe the samples (L: 349-351).
- Line 360: So, what was the name of the internal standards used
Response: There is no internal standard. Here, each sample extract was mixed as the quality control samples (QC) to monitor deviations of the analytical results and evaluate potential errors from analytical instruments. We have added the sentence to describe the samples (L: 351-352).
- Line 370: Also look at what is required in LC-MS. Look through this and add some of these points to your manuscript if needed.
Response: Thanks for your suggestion and we re-described the LC-MS method (L: 349-362).
- Line 373: So where is the Mass spectrums of identified metabolites?
Response: The mass spectrums has been added (see Figure S6).
- Line 525: Check current reviews and research on ginkgo TTL and add them.
Are you telling me that there is no research on gingko TTL in 2023?
Response: We have added the latest published relevant papers (L: 543-545).
Reviewer 2 Report (New Reviewer)
Comments and Suggestions for Authors
Comments for the manuscript entitled: "Inferring the regulatory network of miRNAs on terpene trilactone biosynthesis affected by environmental conditions" submitted by Ying Guo et al.:
This study is very complex, with a lot data that represented NCBI BioProject number: PRJNA649066.
The subject is very interesting, aiming to elucidate (largely) the molecular mechanism regulating the biosynthesis of terpene trilactones from leaves (TTL) of Ginkgo biloba L. grown in three areas of China (each with a different environment) and the interference of this mechanism with environmental conditions. It also aims to strengthen the scientific basis for optimizing the medicinal value of ginkgo leaves. The present study focused RNA-seq and small RNS-seq high-throughput sequencing data, correlating them with metabolomics data from Ginkgo biloba leaves.
The proposed objectives were demonstrated by the results obtained on: accumulation of TTL in response to environmental changes, identification of miRNAs responsible for the regulation of TTL metabolism, and revealing the regulatory mechanism mediated by miRNAs in response to TTL metabolism in different environments.
My comments are below:
1. In line 282, Catharanthus roseus should be written in italics. You probably missed it.
2. In line 327 you mentioned that the daily climate data of the three test sites in the month of sampling were taken from https://data.cma.cn/. This source should also be mentioned at "Refeences", after reference 57..
3. In lines 387-389 all https://www ... in brackets should be mentioned at "References" after reference 57.
I wish you success in publishing this study!
Author Response
RESPONSE TO REVIEWERS
Reviewer #2:
The subject is very interesting, aiming to elucidate (largely) the molecular mechanism regulating the biosynthesis of terpene trilactones from leaves (TTL) of Ginkgo biloba L. grown in three areas of China (each with a different environment) and the interference of this mechanism with environmental conditions. It also aims to strengthen the scientific basis for optimizing the medicinal value of ginkgo leaves. The present study focused RNA-seq and small RNS-seq high-throughput sequencing data, correlating them with metabolomics data from Ginkgo biloba leaves.
The proposed objectives were demonstrated by the results obtained on: accumulation of TTL in response to environmental changes, identification of miRNAs responsible for the regulation of TTL metabolism, and revealing the regulatory mechanism mediated by miRNAs in response to TTL metabolism in different environments.
My comments are below:
- In line 282, Catharanthus roseus should be written in italics. You probably missed it.
Response: Thanks for your suggestions. Corrected (L: 269).
- In line 327 you mentioned that the daily climate data of the three test sites in the month of sampling were taken from https://data.cma.cn/. This source should also be mentioned at "References", after reference 57.
Response: According to the suggestions, we have added corresponding references (see reference 40).
- In lines 387-389 all https://www ... in brackets should be mentioned at "References" after reference 57.
Response: According to the suggestions, we have added corresponding references (see reference 42, 46, and 47).
Round 2
Reviewer 1 Report (Previous Reviewer 2)
Comments and Suggestions for Authors
The authors have answered my comments.
Thanks
Isaac Duah Boateng
Comments on the Quality of English Languageminor revision
Author Response
Dear Isaac Duah Boateng,
Thank you very much for your kindly advice, and we have considered all the suggestions.
Sincerely,
Ying Guo
This manuscript is a resubmission of an earlier submission. The following is a list of the peer review reports and author responses from that submission.
Round 1
Reviewer 1 Report
Comments and Suggestions for Authors
In the manuscript the authors reported the characterization by metabolic, RNAseq and smallRNAseq approaches of leaf of Ginkgo biloba collected in 3 different environments in order to understand the regulatory network of terpene trefoil lactones biosynthesis.
In the manuscript, the genome-wide expression analysis techniques used provides only a descriptive analysis of the effect of the environment on the production of TTLs without any mechanistic explanation and without safely identifying any gene or miRNA involved in the biosynthesis of these compounds. It is a purely bioinformatics work without any biological validation.
Furthermore, sequencing data were presented in a largely unsatisfactory way.
The authors provide a generic characterization of the 3 environments. They must report all climatic data (daily or weekly) of the 3 environments in the months involved in the experiment (from planting to sampling).
Were the same plant samples used for RNAseq, smallRNAseq and metabolome?
Please report expression values of all 9 biological replicates for each gene and statistical analyzes to determine the reliability of the DEGs.
Please validate the expression of a suitable number of genes by RT-qPCR.
For small RNAs, report the expression values of all miRNAs in all 9 biological replicates, the miRNA and miRNAs sequences of all novel miRNAs, the precursors of the novel miRNAs including the secondary structure. Verify that they are novels and not variants of other miRNAs already identified.
Please report for all miRNAs the putative target genes identified by psRNATarget. However, target prediction by bioinformatics methods is highly unsatisfactory and unreliable. If the authors want to do a correlation analysis between miRNA, gene expression and metabolites, identify the target genes using Degradome analysis.
Please validate the expression of a suitable number of miRNAs by RT-qPCR or northern analysis.
Reviewer 2 Report
Comments and Suggestions for Authors
I have carefully read the article. I found this topic exciting, but I have some comments that I have attached as "a word file" that you need to download and use to improve the manuscript. A lot of grammatical errors. I have corrected some grammatical errors and highlighted the ones I have corrected as " red color" in the attached document. I am willing to review this article again after you incorporate my comments. Attached are my comments

Round 2
Reviewer 1 Report
Comments and Suggestions for Authors
The authors have only partially responded to my suggestions and the improvements have been minimal and not sufficient. Validations of RNA-seq and smallRNA-seq data are not needed to validate the technique itself, but to validate the authors' bioinformatic analyses. They cannot expect me or a reader to repeat all the analyzes to verify that the authors have not made mistakes. The tagets have been indicated with a code that means nothing. Tatget annotations? the degree of reliability of the analysis? Analyzes with degradame or some target validation system are not optional, but necessary.
Reviewer 2 Report
Comments and Suggestions for Authors
Round 3
Reviewer 1 Report
Comments and Suggestions for Authors
The authors continue to respond minimally to requests and have not added any required biological validation: mRNA/miRNA (RT-qPCR) and target (RACE or Degradome). They simply re-analyze the bioinformatic data. In table S1 they never added the gene annotation, is it necessary to ask for it explicitly? after the same thing was asked for table S3?
Reviewer 2 Report
Comments and Suggestions for Authors
The authors have answered all my questions. Thus, the article should be accepted for publication following this minor revision.
1. A lot of grammatical errors. I have corrected some of them and highlighted them in blue in the attached document. Consider reading through it thoroughly to correct some of the grammatical errors.
2. Regarding the effect of drying temperature on ginkgolides and bilobalide, There have been works of literature that have shown that drying methods and temperature affect the ginkgolide A, B, and C content of ginkgo biloba seed.
Boateng, I. D., & Yang, X. (2021). Do nonthermal pretreatments followed by intermediate-wave infrared drying affect toxicity, allergenicity, bioactive, functional groups, and flavor components of Ginkgo biloba seed ? A case study. Industrial Crops & Products, 165, 113421. https://doi.org/10.1016/j.indcrop.2021.113421
Boateng, I. D., & Yang, X. (2022). Ginkgo biloba L . seed ; A comprehensive review of bioactives, toxicants, and processing effects. Industrial Crops & Products, 176, 114281. https://doi.org/10.1016/j.indcrop.2021.114281
Boateng, I. D. (2022). A critical review of current technologies used to reduce ginkgotoxin, ginkgotoxin-5'-glucoside, ginkgolic acid, allergic glycoprotein, and cyanide in Ginkgo biloba L. seed. Food Chemistry. https://doi.org/10.1016/j.foodchem.2022.132408
Boateng, I. D., & Yang, X. (2022). Ginkgo biloba L . seed ; A comprehensive review of bioactives, toxicants, and processing effects. Industrial Crops & Products, 176, 114281. https://doi.org/10.1016/j.indcrop.2021.114281
Well, it can be argued that this was on gingko seeds and not on ginkgo leaves. However, the content of TTL in ginkgo leaves is far higher than in ginkgo seeds.
Liu, X. G., Lu, X., Gao, W., Li, P., & Yang, H. (2022). Structure, synthesis, biosynthesis, and activity of the characteristic compounds from Ginkgo biloba L. Natural Product Reports, 39(3), 474-511.
So if you are saying there was no significant difference at 70C for 48 hrs, it might be acceptable or debatable depending on the drying method and conditions. All the same, research creates gaps for future experiments.
Thanks
